

# Identification of immune-related genes in acute myocardial infarction based on integrated bioinformatical methods and experimental verification

Jian Liu[1], Lu Chen[1], Xiang Zheng[2] and Caixia Guo[1]

[1] Cardiovascular Center, Beijing Tongren Hospital, Capital Medical University, Beijing, China
[2] Urology, Beijing Chaoyang Hospital, Capital Medical University, Beijing, China

## ABSTRACT

**Background**. Acute myocardial infarction (AMI) is one of the leading causes of death worldwide. The etiology of AMI is complex and has not been fully defined. In recent years, the role of immune response in the development, progression and prognosis of AMI has received increasing attention. The aim of this study was to identify key genes associated with the immune response in AMI and to analyze their immune infiltration.

**Methods**. The study included a total of two GEO databases, containing 83 patients with AMI and 54 healthy individuals. We used the linear model of microarray data (limma) package to find the differentially expressed genes associated with AMI, performing weighted gene co-expression analysis (WGCNA) to further identify the genes associated with inflammatory response to AMI. We found the final hub genes through the protein-protein interaction (PPI) network and least absolute shrinkage and selection operator (LASSO) regression model. To verify the above conclusions, we constructed mice AMI model, extracting myocardial tissue to perform qRT-PCR. Furthermore, the CIBERSORT tool for immune cells infiltration analysis was also carried out.

**Results**. A total of 5,425 significant up-regulated and 2,126 down-regulated genes were found in GSE66360 and GSE24519. A total of 116 immune-related genes in close association with AMI were screened by WGCNA analysis. These genes were mostly clustered in the immune response on the basis of GO and KEGG enrichment. With construction of PPI network and LASSO regression analysis, this research found three hub genes (SOCS2, FFAR2, MYO10) among these differentially expressed genes. The immune cell infiltration results revealed that significant differences could be found on T cells CD4 memory activated, Tregs (regulatory T cells), macrophages M2, neutrophils, T cells CD8, T cells CD4 naive, eosinophils between controls and AMI patients.

## INTRODUCTION

Acute myocardial infarction (AMI) is the most serious form of coronary heart disease (CHD) and poses a serious threat to human health (*Chaitman et al., 2021*). The biomarkers of myocardial injury such as creatine kinase-MB (CK-MB), cardiac troponin T (cTnT)

Corresponding author
Caixia Guo, ccxgbb@163.com

and cardiac troponin T (cTnI) are important diagnostic indicators of acute myocardial infarction (*Gulati et al., 2021*). However, these biomarkers remain virtually unchanged for a long time before the onset of myocardial infarction. Traditional markers of myocardial injury only reflect the degree of cardiac injury, not the pathogenesis and prognosis of AMI.

In recent years, immune response has attracted more and more attention in the development of CHD. According to the current view, CHD is an inflammatory process based on atherosclerosis, which is characterized by lipid accumulation, immune cell infiltration and fibrous fat plaque formation (*Libby, 2021*). AMI is the most serious kind of CHD. The unstable plaque in the blood vessels of patients is more prone to rupture, and the ruptured plaque will gather a large number of platelets, causing thrombosis in the blood vessels, and the ischemic myocardium begins to damage and die (*Damluji et al., 2021*). After myocardial infarction, myocardial infarction cells are recognized, swallowed and cleared by the immune system, then scar tissue is formed (*Joshi et al., 2018*). Appropriate inflammatory reaction reduces the infarct area, promoting the formation of scar in the infarct area, thus contributing to the recovery of ischemic myocardium (*Westman et al., 2016*). Excessive inflammatory reaction causes apoptosis of myocardial cells in non-infarct area, hypertrophy and fibrosis of myocardial tissue, leading to pathological reconstruction of ischemia related tissues and myocardial dysfunction (*Mahtta et al., 2020*; *Ong et al., 2018*). Therefore, regulating the expression of immune response and inflammatory factors is of great significance for the prevention, treatment and prognosis of myocardial infarction.

Unfortunately, the mechanisms of inflammatory regulation in AMI are not well understood and not enough studies have been conducted in this area (*Kilgallen et al., 2022*). Firstly, many inflammatory factors exhibit both pro-myocardial injury and myocardial protective effects. Such as IL-6, an inflammatory factor traditionally considered as a pro-inflammatory factor that promotes inflammatory progression under various pathological conditions (*Qu et al., 2014*). But its anti-inflammatory properties in cardiovascular disease have been increasingly recognized (*Scheller et al., 2011*). Similarly, TNF-$\alpha$, traditionally considered a pro-inflammatory factor, can also exhibit anti-inflammatory properties. Its receptor-TNFR2 has been shown to have anti-inflammatory and histoprotective properties in encephalomyelitis, colitis, heart failure, inflammatory arthritis, and myocardial infarction (*Blüml et al., 2012*). These opposing effects suggest that the identified inflammatory factors and unspecified inflammatory factors may be somehow related. Secondly, in clinical practice, targeting inflammatory factors therapy has been applied. Although its status is increasingly important, its efficacy is still lacking to a considerable extent. Probably because the relations between various inflammatory factors are rather complicated (*Rurik, Aghajanian & Epstein, 2021*). Newly discovered inflammatory factors may reveal the reasons for the poor efficacy of certain inflammatory therapies. In summary, it is necessary to search for new immune factors associated with AMI. Finding the relations with other known inflammatory factors mechanisms, assess prognosis against these immune factors and immune cells, and develop targeted therapies to provide new clinical options to benefit patients.

This study analyzed the gene expression profiles of AMI patients and healthy controls in the public database of AMI. To diagnose the AMI patients in early phrase, we also

established a least absolute shrinkage and selection operator (LASSO) model to predict the occurrence of AMI, and verified it with another dataset and animal AMI model. The purpose of this study is to identify the early diagnosis molecules of the disease and improve the prognosis of patients with AMI.

# MATERIALS AND METHODS

## Microarray data

In order to find the differentially expressed genes (DEGs) related to immune response in AMI, this study utilized two datasets GSE24519 and GSE66360 in the National Biotechnology Information Center (NCBI) database (Gene Expression Omnibus, https://www.ncbi.nlm.nih.gov/geo/) These two datasets included 83 patients with AMI and 54 healthy individuals, containing the expression of more than 40,000 genes. Limma package in R (http://www.r-project.org) was used for background correction and standardization of gene expression. Normalization between arrays was performed using the quantile algorithm with limma package (*Ritchie et al., 2015*). The entire analysis process of this study was shown in Fig. 1.

## Select differentially expressed genes

Among the genes detected in transcriptome sequencing, we used the limma package in R to analyze the differentially expressed immune related genes between AMI group and normal samples. For the threshold of DEGs, we set it as $|logFC|>1$ and $P < 0.01$ (*Li, 2012*). We applied volcano map, Venn diagram and heatmap through R to show the differentially expressed genes in the two datasets.

## Construction of weighted gene co- expression network analysis (WGCNA)

In order to identify immune regulatory genes related to AMI, we applied weighted gene co-expression network analysis (WGCNA) to explore the characteristics of related gene networks (*Yang et al., 2021*). WGCNA can effectively integrate gene expression and trait data from different samples, identifying highly synergistic gene sets and functional pathways. We divided the found differential genes into different modules through the WGCNA package in R, determining the module with the highest correlation with AMI, and finally selected the genes related to immune response.

## Functional enrichment analysis of DEGs

Use the constructed gene annotation database Gene Ontology (GO, http://geneontology.org/), Kyoto Encyclopedia of Genes and Genomes (KEGG, https://www.kegg.jp/) to get the enrichment analysis results. GO is a practical database resource for understanding high-order functions and biological systems (cells, organisms and ecosystems) from molecular level information, such as genome sequencing results generated by high-throughput experimental technology, through specific algorithms. Cluster the genes according to the knowledge of gene annotation database, and remove the redundant results to obtain the final gene enrichment (*The Gene Ontology Consortium, 2019*). KEGG analysis usually inputs

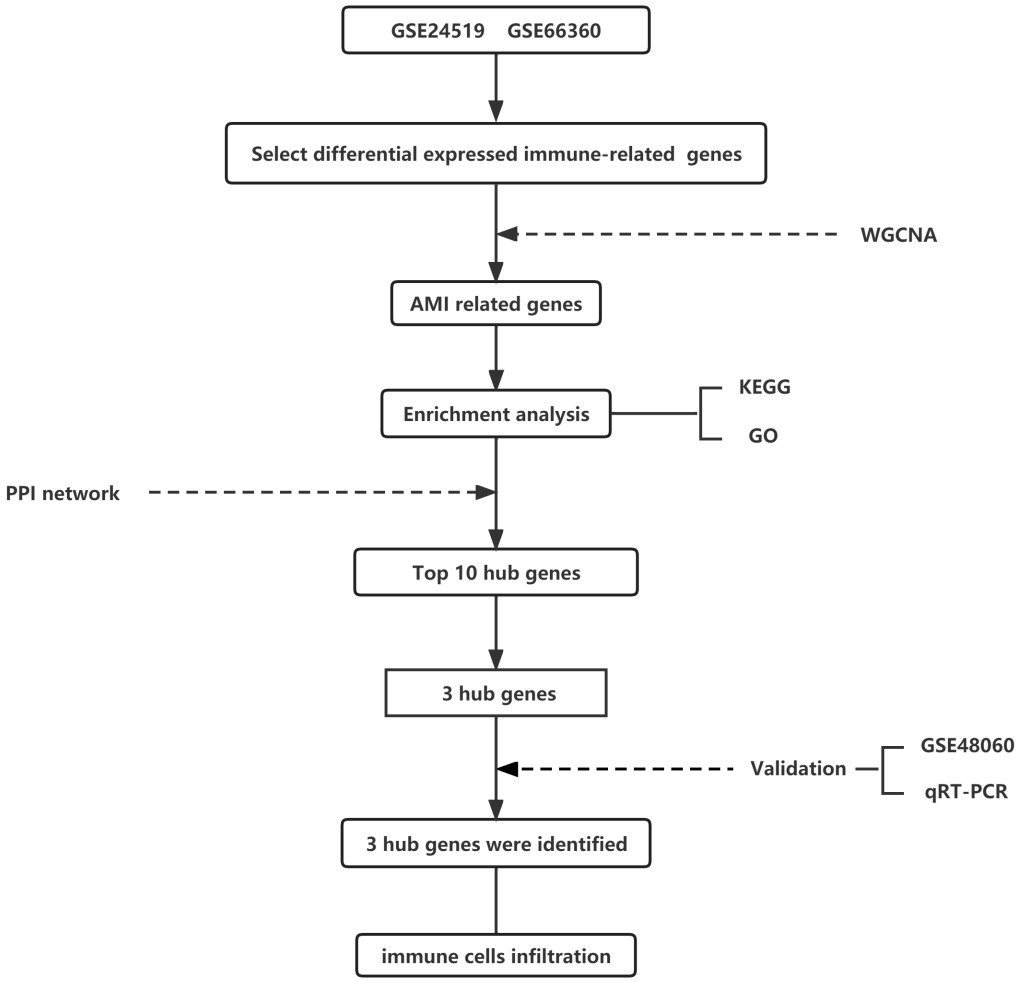

**Figure 1** Flow chart to illustrate the analysis process of the present study.

gene sets with obvious differences in expression in the data, calculating the hypergeometric distribution relationship between these different genes and some specific branches in KEGG classification, and then returns a *P* value for each item with different genes. A smaller *P* value indicates that the different genes are enriched in KEGG (*Yu et al., 2012*). In this way, we can preliminarily understand which pathways in the disease group have changed significantly compared with the control group.

## Construction of PPI network

The protein-protein interaction (PPI) analysis is crucial in interpreting AMI pathogenesis in terms of molecular mechanisms of the important cellular activities. The construction of PPI network is as described previously (*Szklarczyk et al., 2019*). Briefly, we imported DEGs to The Search Tool for the Retrieval of Interacting Genes database (STRING, http://stringdb.org), which was used to predict potential interactions among mRNAs. Based on the analysis results, we visualized it and explored the hub genes by Cytoscape 3.8.2.

Then, based on topology, the region with most dense connections in the PPI network were detected through the MCODE plugin and Cytohubba which may aid in identifying hub genes (*Shannon et al., 2003*).

## Mice acute myocardial infarction model

All animal experiments were reviewed by the institutions of Capital Medical University and approved by the ethics committee, and were carried out in accordance with international ARRIVE (Animal Research: Reporting of In Vivo Experiments) guidelines for animal experiments. We obtained 12 male C57BL/6 mice (20–22 g; 6 weeks; $n = 6$ of each group) from Vital River (USA). The approval number was AEEI-2020-107. All mice were kept in individually ventilated cages at a temperature of 20 °C to 24 °C, humidity of 50% to 60%, 60 air exchanges per hour in the cages, and a 12/12-hour light/dark cycle with the lights on at 5:30 AM.

The mice were randomly divided into two groups and underwent surgery for acute myocardial infarction and sham operation, as previously described (*Liu et al., 2021*). Briefly, mice were anesthetized with 2 L/min inhalation of 2% isoflurane and 100% $O_2$. The mouse heart was carefully removed from the fourth costal space, and the anterior descending branch of the left coronary artery was ligated with 6-0 silk thread, causing myocardial ischemia. After ligation, the heart was sent back to the chest, and the muscles and skin were sutured. Sham operated mice only underwent thoracotomy without ligation of the coronary artery. 24 h after surgery, mice were intraperitoneally injected with an excess of pentobarbital (100 mg/kg), and then cervical vertebrae were dislocated. These hearts were collected and stored in liquid nitrogen for further study. This mice MI model has been skillfully applied in our research group.

## Judgment of mice AMI model

After establishing the AMI model, we stained mice with 2, 3, 5-triphenyltetrazolium chloride (TTC) to determine the success of the model. TTC (Sigma-Aldrich, St. Louis, MO, USA) was dissolved in phosphate-buffered saline (PBS) and injected into the coronary artery through the end of the aortic arch. The hearts were incubated in 1% TTC solution at 37 °C for 5 min and then fixed in 4% paraformaldehyde for 4 min. The heart was then frozen at −20 °C for 5 min, transected in the coronal plane, and photographed under a body view microscope. Photographs were taken under a stereomicroscope. The white area represents infarcted heart tissue, while the red area represents non-infarcted tissue.

## Quantitative reverse transcription (qRT-PCR)

Tissue RNA was extracted from mice hearts with TRIzol reagent (Invitrogen, Waltham, MA, USA). The concentrations and purity of RNA were determined by measuring the absorbance at 260 nm and 260/280 absorbance ratio (Eppendorf, Hamburg, Germany). A total of 1 μg of total RNAs were reverse-transcribed to generate first-strand cDNA; 1 μl of forward primer, 1 μl of reverse primer, 10 μl of SYBR Premix Ex Taq (Takara, Shiga, Japan) and 6 μl of DNA-free water. The 7500 Real-Time PCR system (Applied Biosystems, Waltham, MA, USA) was used to perform the experiments. The primers employed for the amplification of individual subunits were in Table 1, the primers sequences were from
**Table 1  Detailed information on primer sequence of each gene symbol and internal reference gene.**

| Gene symbol | Primer Sequence |
|---|---|
| SOCS2 | Forward: 5′-TTTGGCGCAGAAAAACTCGG-3′ |
| | Reverse: 5′-GTTCCCCGGGTGACGTTTAT-3′ |
| FFAR2 | Forward: 5′-GGAAACGGGAAGCCTCGTTC-3′ |
| | Reverse: 5′-CCAGTCTGGGGTCATTCTCC-3′ |
| MYO10 | Forward: 5′-TCCATCTCTGTGGAGGCAGA-3′ |
| | Reverse: 5′-AGACCAGCCTTGGGTTTCAC-3′ |
| $\beta$-ACTIN | Forward: 5′-TGAGCTGCGTTTTACACCCT-3′ |
| | Reverse: 5′-GCCTTCACCGTTCCAGTTTT-3′ |

NCBI (https://www.ncbi.nlm.nih.gov/gene/), the synthesis of primers were completed by Sangon Biotech (Shanghai, China) (Zhang et al., 2022a).

## Statistical analysis

All the data processing and statistical analysis were performed in R software (version 4.0.5), Cytoscape (version 3.8.2), SPSS 22.0, GraphPad Prism 8 (GraphPad Software, USA), Gene Ontology (GO) analysis and Kyoto Encyclopedia of Genes and Genomes (KEGG) pathway analysis were applied in The Database for Annotation, Visualization and Integrated Discovery (DAVID). Experimental data are presented as means $\pm$SEM, differences between groups were assessed by independent $t$-test. the $P$ value less than 0.05 at two-group was considered statistical significance.

## RESULTS

### Identification of differentially expressed mRNAs

In order to clarify the expression of immune response related genes in AMI, this study examined the array data between AMI patients and healthy individuals from GSE24519 and GSE66360. Subsequently, limma package in R was used to identify and analyze the differentially expressed genes in the transcriptome sequencing in the datasets. We defined the fold change greater than 2 times as differential expression, and the $P <$ 0.01. In this discovery, there were 5,831 genes in GSE24519, including 4,207 up-regulated genes and 1,624 down-regulated genes (Fig. 2A). There were 1,720 genes in GSE66360, including 1,218 up-regulated genes and 502 down-regulated genes (Fig. 2B). These two datasets contained a total of 7,551 eligible differentially expressed mRNAs (5,425 up-regulated genes and 2,126 down-regulated genes), and 445 mutual genes (397 up-regulated genes and 48 down-regulated genes) (Figs. 2C–2E). Figures 2F–2G showed the expression levels of the top 10 up-regulated and top 10 down-regulated genes in each dataset.

### Co-expression network of immune-related genes

To further screen the DEGs, the co-expression network of DEGs in GSE24519 and GSE66360 was constructed by weighted gene co-expression network analysis (WGCNA). The WGCNA package in R was used to cluster genes into different modules, and each module has a unique color. The connection between genes was determined and weighted

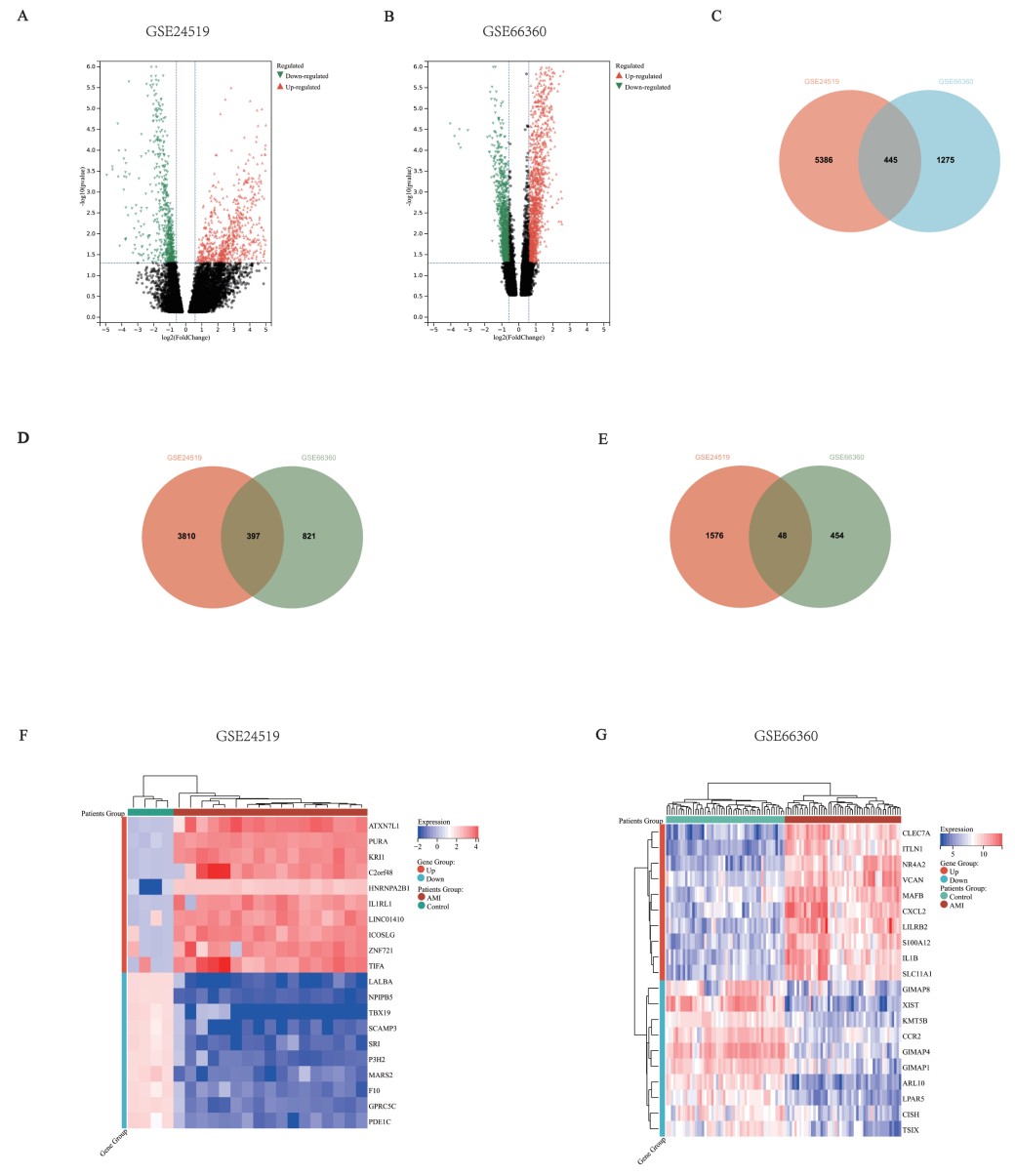

**Figure 2   Common mRNA expression patterns in the AMI patients and healthy individuals.** (A–B) mRNA expression volcano map, different colors represented different expressions. Each spot represented a gene, the red dots meant up-regulated genes, the green dots meant down-regulated genes. (A) GSE24519. There were 5,831 differentially expressed mRNA matching the criteria, including 4,207 up-regulated and 1,624 down-regulated. (B) GSE66360. There were 1,720 differentially expressed mRNA matching the criteria, including 1,218 up-regulated and 502 down-regulated. (C) Venn Diagram of total DEGs in the two datasets, there were 5,831 genes in GSE24519, 1,720 genes in GSE66360, including 445 mutual genes. (D) Venn Diagram of up-regulated DEGs in the two datasets, 4,207 genes in GSE24519, 1,218 genes in GSE66360, including 397 mutual genes. (E) Venn Diagram of down-regulated DEGs in the two datasets, 1,624 genes in GSE24519, 502 genes in GSE66360, including 48 mutual genes. (F–G) Heatmap showed the top 10 up and down regulated genes in total samples. (F) GSE24519 (G) GSE66360.

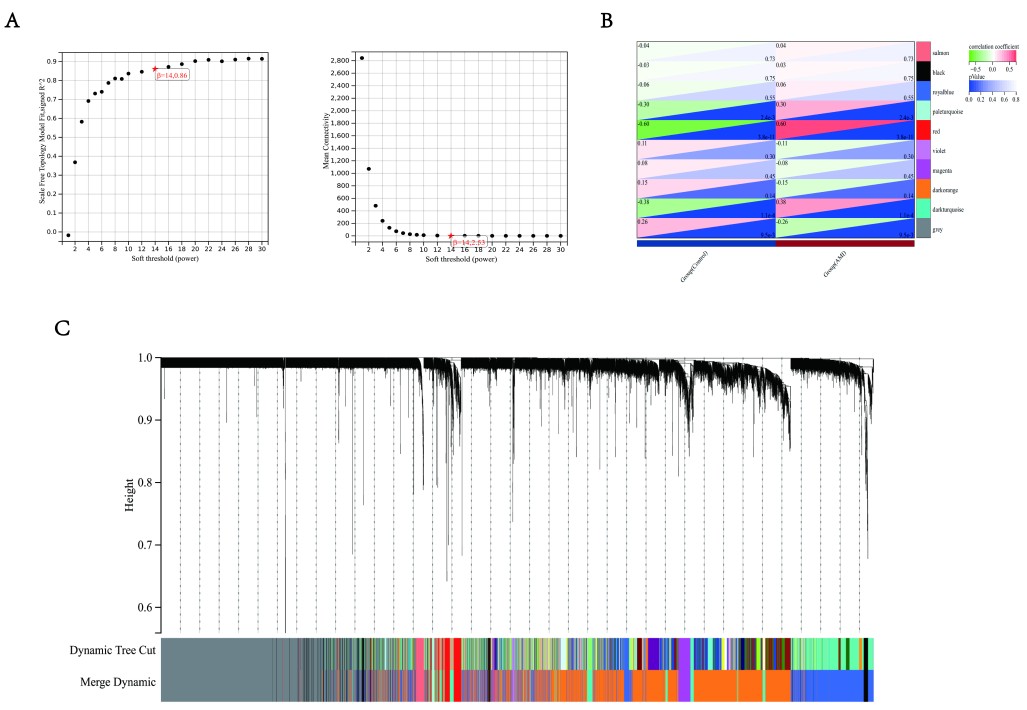

**Figure 3  Weighted gene co-expression analysis (WGCNA) based on the immune-related DEGs.** (A) Analysis of network topology for various soft-thresholding powers. The left panel showed the scale-free fit index (*y*-axis) as a function of the soft-thresholding power (*x*-axis). The right panel showed the mean connectivity (*y*-axis) as a function of the soft-thresholding power (*x*-axis). (B) Module-trait relationship: each row corresponded to a module eigengene and each column to a trait. Each cell contained the corresponding correlation and *p*-value. The table was color-coded by correlation according to the color legend. Distribution of average gene significance and errors in the modules were associated with AMI patients and controls. (C) Gene dendrogram and module colors, the row underneath the dendrogram showed the module assignment determined by the Dynamic Tree Cut.

according to the expression level of related genes in different samples (Fig. 3A). Further, the connection matrix is transformed into topological overlap matrix (TOM) to detect the gene commonality in the network. Then, hierarchical clustering and covariance coefficient were used to identify the clustering of highly correlated gene, that is, the final module. The minimum size of the gene tree is 30. The power value is selected to define the high positive correlation between genes in the same module. Among these modules, the genes in the red module have the largest association with the occurrence of AMI, with the largest coefficient values ($r = 0.6$, $P < 0.01$ (Fig. 3B). This module belongs to up regulated genes, with a total of 132 genes (Fig. 3C). We compared the immune gene list and removed 16 genes that were not immune related (*Cao et al., 2020*). Herein, 116 immune-related genes in close association with AMI were screened by WGCNA analysis.

## GO and KEGG pathway enrichment analyses

In order to clarify which biological processes the above 116 immune-related genes in the module participate in, this study used Gene Ontology (GO) and Kyoto Encyclopedia of

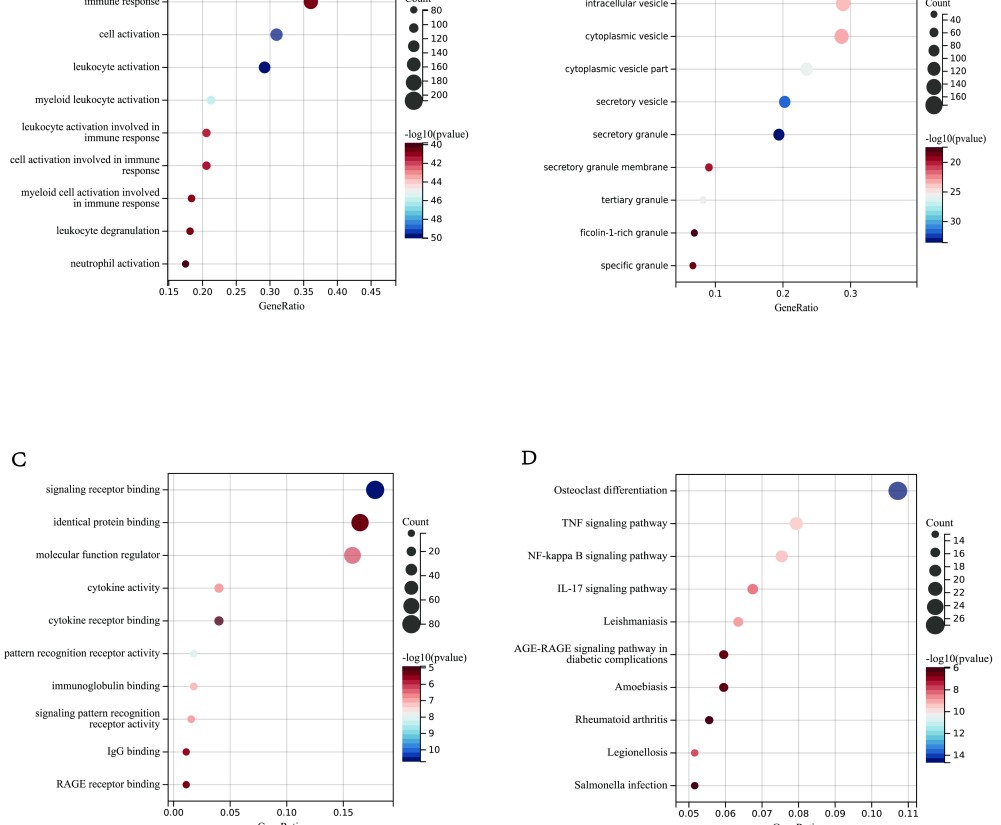

**Figure 4 Enrichment Analyses outcomes.** (A–D) The bar graphs of Gene Ontology annotation and Kyoto Encyclopedia of Genes and Genomes pathway enrichment of DEGs included BP (biological process), CC (cellular component), MF (molecular function), KEGG (signaling pathway); (A) BP (B) CC (C) MF (D) KEGG.

Genes and Genomes (KEGG) for analysis. We used the clusterprofiler package in R, and $P < 0.01$ was considered statistically significant. The enrichment analysis results showed 30 GO terms, including the top 10 biological processes (BPs), top 10 cellular components (CCs) and top 10 molecular functions (MFs). The results showed that the DEGs in BPs was mainly enriched in neutrophil activation and neutrophil degranulation (Fig. 4A). In CCs was mainly enriched in the third granule, secretory granule membrane and specific granules (Fig. 4B). In MFs was mainly enriched in pattern recognition receptor activity, immunoglobulin binding, and signal pattern recognition receptor activity (Fig. 4C). In addition, KEGG showed the following approaches were obtained in this study: NF-KB signal pathway, TNF signal pathway, IL-17 signal pathway (Fig. 4D). These processes and pathways were all related to immune response.

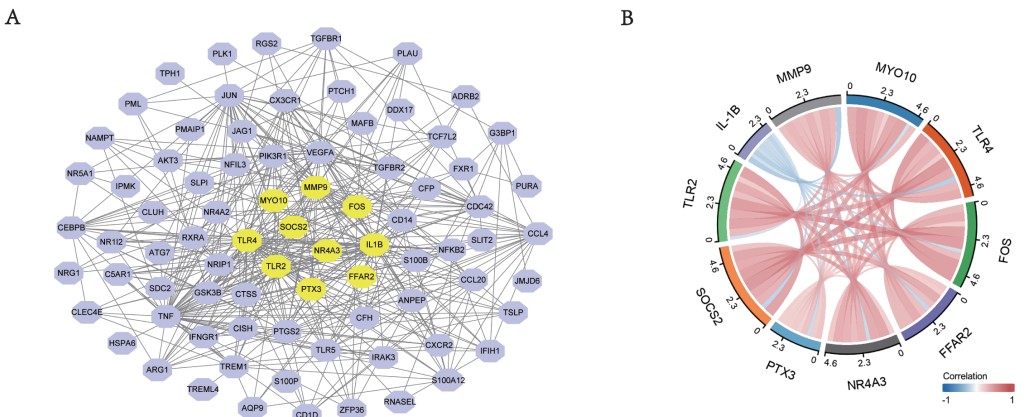

**Figure 5   The PPI networks of top DEGs.** (A) All the circles were proteins encoded by top DEGs. Purple color represented the genes that met the criteria, yellow color represented the top 10 most significant genes. The black lines represented the relationship between two genes. (B) Chord diagram of 10 hub genes. The solid lines meant the relationship of every two genes, and the depth of color represented the closeness of their connection.

## Construction of PPI network and hub genes identification

The construction of protein-protein interaction (PPI) network was to elucidate DEG interaction. Data analysis was based on STRING database. If the interactive pair score was higher than 0.4, it would be visualized in Cytoscape (Fig. 5A). The identification of hub genes in the PPI network was done using the CytoHubba plugin in Cytoscape. We selected the top 10 hub genes, which were IL-1B, MMP9, MYO10, TLR2, TLR4, FOS, FFAR2, NR4A3, PTX3, SOCS2. All of these genes were up-regulated. Figure 5B showed the relationships among these 10 hub genes, and it could be seen these genes were all closely linked.

## Results of immune cell infiltration

It is because of the possible immune response in AMI, which makes it necessary to explore possible immune cells involved in the progression of AMI. Dataset GSE24519 provides expression data from peripheral blood of patients with AMI that has the potential to achieve this goal. We used CIBERSORT algorithm to calculate the infiltration of immune cells in AMI group and normal control group. The results (Fig. 6A) showed the proportion of T cells CD4 memory activated ($P < 0.01$), Tregs (Regulatory T cells) ($P < 0.001$), Macrophages M2 ($P < 0.01$), Neutrophils, ($P < 0.01$) was much higher in AMI group than in control group. T cells CD8 ($P < 0.01$), T cells CD4 naive ($P < 0.01$), Eosinophils ($P < 0.01$) was much lower in AMI group than in control group. Moreover, the correlation of 22 types of immune cells were also calculated, the results (Fig. 6B) revealed that naive T cells CD8 were significantly positively correlated with B cells naive ($r = 0.2$, $P < 0.01$), negatively correlated with B cells memory ($r = -0.21$, $P < 0.001$); T cells CD4 naive were significantly positively correlated with Tregs ($r = 0.25$, $P < 0.001$), negatively correlated with NK cells resting ($r = -0.24$, $P < 0.001$); T cells CD4 memory resting were significantly
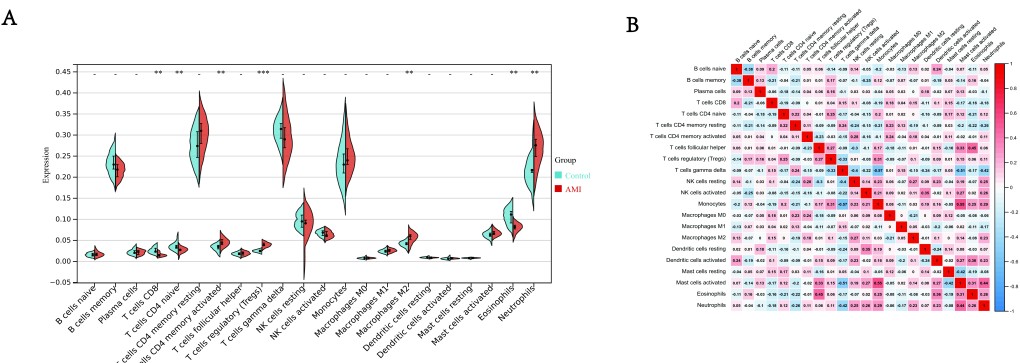

**Figure 6 Immune infiltration analysis.** (A) Split violin plot of the estimated proportion of 22 types of immune cells between control and AMI using the dataset of GSE24519. (B) Correlation heat map of 22 types of immune cells. Positive and negative correlations were respectively shown in blue and red color, whereas the number represented the correlation parameters. **$P < 0.01$, ***$P < 0.001$ compared with control for two groups.

positively correlated with T cells gamma delta ($r = 0.24$, $P < 0.01$), negatively correlated with Neutrophils (r $=-0.26$, $P < 0.001$); T cells CD4 memory activated were significantly positively correlated with NK cells resting ($r = 0.28$, $P < 0.001$), negatively correlated with T cells follicular helper (r $=-0.23$, $P < 0.001$); Tregs were significantly positively correlated with Monocytes ($r = 0.31$, $P < 0.001$), negatively correlated with T cells gamma delta (r $=-0.33$, $P < 0.001$); Monocytes were significantly positively correlated with mast cells activated ($r = 0.55$, $P < 0.01$), negatively correlated with T cells gamma delta (r $=-0.57$, $P < 0.001$); Macrophages M1 were significantly positively correlated with T cells gamma delta ($r = 0.15$, $P < 0.001$), negatively correlated with dendritic cells activated (r $=-0.2$, $P < 0.001$); Macrophages M2 were significantly positively correlated with NK cells resting ($r = 0.27$, $P < 0.001$), negatively correlated with Macrophages M0 (r $=-0.21$, $P < 0.001$); Eosinophils were significantly positively correlated with T cells follicular helper ($r = 0.45$, $P < 0.001$), negatively correlated with T cells CD4 resting memory ($r = -0.22$, $P < 0.001$); Neutrophils significantly positively correlated with mast cells activated ($r = 0.44$, $P < 0.001$), negatively correlated with T cells gamma delta ($r = -0.42$, $P < 0.001$). The results in Fig. 6B showed a potential relationship between different immune cells, help us discover more mechanisms of immune infiltration in AMI.

## Establishment of the diagnostic model based on LASSO

We built a least absolute shrinkage and selection operator (LASSO) prediction model of disease occurrence by constructing a penalty function to obtain a more refined model. In this study, whether AMI occurred was used as the end-point event, and different amounts of gene expression in each sample were used as variables. First to determine the best model λ value (Fig. 7A), and then verify the top 10 most significant genes found in the PPI network. The results show that the optimum λ value is 0.13. At this time, the coefficient of SOCS2, FFAR2, MYO19 and MMP9 was not zero (Fig. 7B), so the hub genes we finally selected were SOCS2, FFAR2, MYO10, MMP9. Meanwhile, the relationship between MMP9 and

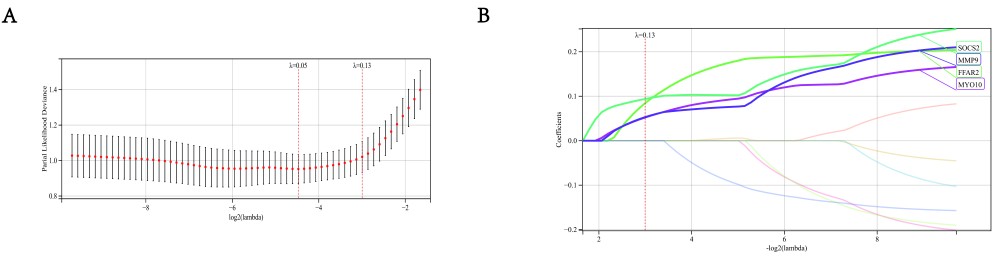

**Figure 7  Establishment of LASSO model.** (A) The gene signature selection of optimal parameter (lambda) in LASSO mode. (B) LASSO coefficient profiles of four differentially expressed genes were selected by the optimal lambda.

AMI is relatively clear, studies between AMI and MMP9 are quite a lot, so we did not take MMP9 as our key research object.

### Validation of the identified hub genes

This research identified these three hub genes related to immune response in AMI, each of them was SOCS2, FFAR2 and MYO10, and then confirmed this conjecture by analyzing the expression of related genes in other samples from GSE48060, which contained 31 patients with AMI and 21 healthy individuals. The expression of these three hub genes were all significantly higher in the AMI group than in the control group, $P < 0.01$ (Fig. 8A). We also performed further animal experiments, constructed mice myocardial infarction model, the arrow pointed the infarction area (Fig. 8C). By measuring the relative expression of RNA in the extracted heart tissue using qRT-PCR, we found that the expression of SOCS2, FFAR2 and MYO10 in the AMI model group was higher than that in the control group, $P < 0.01$, which was statistically significant (Fig. 8B). The above results further verified our previous conjecture.

### Immune cell infiltration

Furthermore, we also explored the correlations between three validated genes and different immune cell types, the results showed that SOCS2 had positive correlation with neutrophils ($r = 0.38$, $P < 0.001$), and negative correlations with NK cells ($r = -0.28$, $P < 0.001$) (Fig. 9A); FFAR2 had positive correlation with Macrophages ($r = 0.59$, $P < 0.0001$), and negative correlations with NK CD56 cells ($r = -0.2$, $P < 0.001$), (Fig. 9B); MYO10 had positive correlation with Th1 cells ($r = 0.66$, $P < 0.0001$), and negative correlations with NK CD56 cells ($r = -0.09$, $P < 0.001$), (Fig. 9C). These results suggested that some types of immune cells were dysregulated in the environment of AMI.

### DISCUSSION

Coronary heart disease (CHD) is one of the most serious reasons that threaten human health, the most serious situation is AMI (*Tsao et al., 2022*). Without proper treatment, the prognosis is poor and the mortality is high (*Bhatt, Lopes & Harrington, 2022*). A large number of studies have shown that immune response plays an important role in the

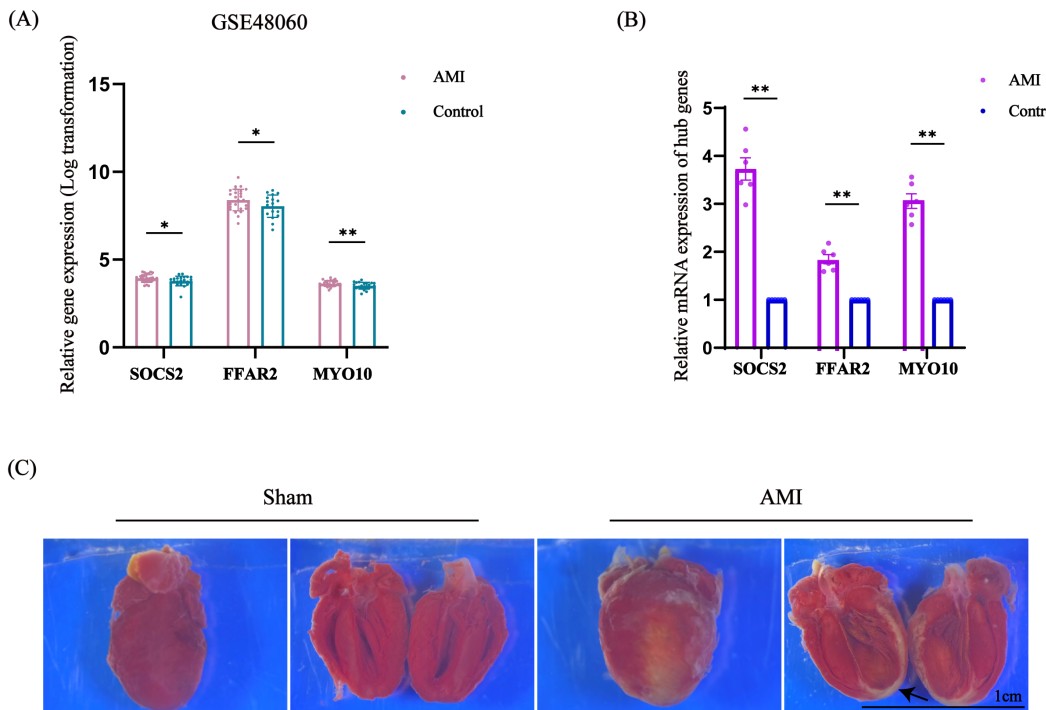

**Figure 8** **Expression of SOCS2, FFAR2, MYO10 in another dataset and the expression levels of SOCS2, FFAR2, MYO10 in mouse myocardial tissue analyzed by qRT-PCR.** (A) The relative expression of SOCS2, FFAR2, MYO10 in GSE48060, the result showed that these genes were higher in AMI group than in control group, $n = 31$ in AMI group, $n = 21$ in control group. (B) qRT-PCR results showed that the expression of SOCS2, FFAR2, MYO10 were higher in AMI group than in the normal one, $n = 6$ per group, the data are expressed as the mean $\pm$ SEM. $^{*}P < 0.05$, $^{**}P < 0.01$ compared with control group. (C) The representative TTC staining images of myocardial infarction area. The black arrow indicates the infarction area.

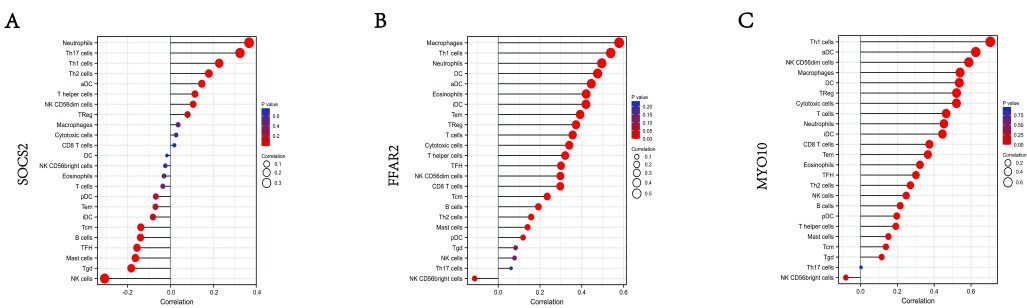

**Figure 9** **Correlation map of 22 types of immune cells and SOCS2, FFAR2, MYO10.** A positive and negative correlation was respectively shown in right and left direction, whereas the high and low *p*-value was respectively shown in light and deep red color. The size of the circle represented the strength of correlation, the larger of the size, the stronger of the correlation. (A) SOCS2. (B) FFAR2. (C) MYO10.

occurrence and development of arteriosclerosis. Autopsy and pathological results showed that there were much inflammatory cells infiltration in the ruptured plaque, indicating a close relationship between immune response and AMI (*Ruparelia et al., 2017*). Immune response is an important part of the whole process from the formation of coronary atherosclerosis to the occurrence of AMI. After myocardial infarction, immune response still plays an important role, affecting the prognosis of patients with AMI (*Afanasieva et al., 2022*). Therefore, it is of great significance to find the key genes that regulate the immune response.

In the present study, we aimed to analyze the relationship between immune response and AMI, as well as the immune infiltration in the disease. A total of 7,551 significantly differentially expressed genes were detected from the two datasets, including 5,425 up-regulated genes and 2,126 down-regulated genes. Using WGCNA analysis, we found 116 immune related genes in red module were highly related to AMI. GO and KEGG enrichment analysis showed that these differentially expressed genes were mainly related to neutrophil activation, pattern recognition receptor activity, immunoglobulin binding, signal pattern recognition receptor activity and other immune responses. Through the construction of PPI network, we found the top 10 hub genes (IL-1B, MMP9, MYO10, TLR2, TLR4, FOS, FFAR2, NR4A3, PTX3, SOCS2), all of which were up-regulated genes. These 10 genes showed the ability to act as biomarkers of AMI. These genes are classical inflammatory factors, all closely associated with the immune response. Through our previous KEGG analysis, these genes were associated with the regulation of IL-17, TNF, and KF-KB (*Pattanaik et al., 2021*). Considering a large proportion of the above the 10 hub genes have been studied in the previous literature, such as IL-1B, MMP9, TLR2, TLR4, FOS, NR4A3, PTX3, which have a more definite association with cardiovascular disease, especially coronary heart disease (*Abbate et al., 2020*; *Cheng et al., 2022*; *Hally et al., 2017*; *Martínez-González et al., 2021*; *Palomer et al., 2020*; *Ristagno et al., 2019*; *Zhu et al., 2022*). Thus, this research will not go through them all here.

We also developed a LASSO disease diagnostic model to further select the genes most associated with AMI. By constructing LASSO model, we selected 4 genes that met the criterion and were of interest to us, which were SOCS2, FFAR2, MYO10 and MMP9. Considering that there were many studies on MMP9 in ischemic heart disease, the relationship between MMP9 and AMI has been studied (*Guo et al., 2021*; *Yabluchanskiy et al., 2013*). Therefore, we did not make MMP9 as our focus and did not involve MMP9 in the subsequent validation. So, we carried out clinical verification through the gene expression of another dataset to verify the expression levels of the three genes (SOCS2, FFAR2, MYO10). We also constructed a mouse myocardial infarction model, extracted tissue RNA, and implemented qRT-PCR for *in vivo* experimental verification. The above results showed that the expression of the 3 candidate genes in the model group was significantly higher than the control group. Therefore, we believed that these 3 hub genes may be important biomarkers to predict the occurrence and development of AMI, and they might play important roles in prognosis of AMI.

The suppressors of cytokine signaling (SOCS) family is composed of a group of proteins. SOCS2 is one of the important components of this family, which is involved in the regulation

of body metabolism, immune response, bone growth, tumorigenesis and other processes (*Val et al., 2020*). SOCS2 can be activated by JAK/STAT pathway, which can be negatively regulated by inhibiting JAK kinase activity (*Slattery et al., 2013*). JAK/STAT pathway plays a wide role in cardiovascular disease. When STAT3 is activated, it can inhibit cardiomyocyte apoptosis, playing a myocardial protective role. It also can increase inflammatory factor IL-1B, IL-6, TNFA and others, accelerating the formation of arterial plaque (*Zhang et al., 2022b*).

Free fatty acid receptor (FFAR2) is a newly discovered specific short chain fatty acid receptor. FFAR2 is involved in metabolism, immune response, cardio-cerebrovascular, as well as cell growth, migration and apoptosis (*Kimura et al., 2020*). FFAR2 is closely related to the activation of inflammatory corpuscles. Inflammatory corpuscle is a multiple protein complex, which can regulate the activation and secretion of pro-inflammatory cytokines such as IL-1, and is a significant player in immune response (*Grundmann et al., 2021*). Moreover, NLRP3 inflammatory corpuscles are closely related to the occurrence of cardiovascular diseases such as atherosclerosis, myocardial infarction, myocardial ischemia-reperfusion and myocardial remodeling after myocardial infarction (*Fujiwara et al., 2018*).

Myosin X (MYO10) is expressed in many tissues of human body. MYO10 participates in the composition of normal cell structure, and plays an important regulatory role in immune cells (*Kerber & Cheney, 2011*). At present, there are many studies in the field of cancer. One study found that in breast tumors, high levels of MYO10 increased the presence of interferon (INF) (*Arjonen et al., 2014*). The close relationship between MYO10 and INF may exist not only in tumor tissues, but also in other tissues. INF is closely related to coronary heart disease, as an important immune inflammatory mediator, produced by helper T cell 1. INF can activate the immune response and further damage the vascular endothelium. It can also activate macrophages and other cells, promote macrophages and vascular endothelial cells to produce IL-1, tumor necrosis factor and platelet-derived growth factor, so that causing formation of atherosclerosis (*Elyasi et al., 2020*).

We found that the mechanisms of action of the three genes mentioned above are related to JAK/STAT, IL-1, and INF-$\gamma$. The upstream or downstream mechanisms of three genes are closely related to the development of AMI as well as its prognosis. Our immune cell infiltration analysis showed a positive association of these three genes with some immune cells, including Th1 cells, neutrophil, macrophage and others. We discovered for the first time that immune regulation of Th1 cells may be able to target FFAR2, SOCS2 and MYO10, providing a theoretical basis for immune-targeted therapy for AMI. In addition, we found that these three genes along with others can affect the levels of other inflammatory factors. For example, all of these three genes can have different levels of effect on IL-6 secretion in lung, neurons (*Kao et al., 2022*; *Sun et al., 2015*; *Xian et al., 2022*). The same effect may also occur in AMI, which needs further study.

In addition, we performed CIBERSORT algorithm to calculate immune infiltration score, the results showed that the proportion of T cells CD4 memory activated, Tregs (Regulatory T cells), Macrophages M2, Neutrophils were much higher in AMI group than control group, T cells CD8, T cells CD4 naive, Eosinophils were much lower in control

group than in AMI group. We also calculated the correlation between immune cells and made a heat map. Therefore, we can formulate new corresponding treatment measures according to the infiltration of these immune cells.

Our current research has several limitations. The number of samples we obtained from the database is relatively small, resulting in some deviations in the analysis of data. In addition, in further research, more blood samples and even tissues of patients are required to verify. In addition, the function and molecular mechanism of genes are very complex, and the prediction based on bioinformatics needs more support from cell and animal experiments.

## CONCLUSIONS

This research identified and verified three up-regulated potential pathogenesis genes (SOCS2, FFAR2, MYO10) related to immune response in AMI. These genes showed a great potential to be used as prediction biomarkers of disease, it may also improve patient prognosis by targeting the mentioned targets to reduce the inflammatory response after AMI. Immune cell infiltration showed possible immune intervention points in the process of AMI.

## ACKNOWLEDGEMENTS

The authors thank the staff of the Department of Pathophysiology, College of Basic Medical Sciences, Capital Medical University for providing experimental techniques to help in this work.

### Funding
This work was supported by the National Natural Science Foundation of China (Grant Nos. 81870265, 82171808) CX. G. The funders had no role in study design, data collection and analysis, decision to publish, or preparation of the manuscript.

### Grant Disclosures
The following grant information was disclosed by the authors:
National Natural Science Foundation of China: 81870265, 82171808.

### Competing Interests
The authors declare there are no competing interests.

### Author Contributions
- Jian Liu conceived and designed the experiments, performed the experiments, analyzed the data, prepared figures and/or tables, authored or reviewed drafts of the article, and approved the final draft.
- Lu Chen conceived and designed the experiments, authored or reviewed drafts of the article, and approved the final draft.
- Xiang Zheng performed the experiments, authored or reviewed drafts of the article, and approved the final draft.
- Caixia Guo conceived and designed the experiments, analyzed the data, prepared figures and/or tables, and approved the final draft.

## Data Availability

The raw measurements are available in the Supplemental Files.

## Supplemental Information

Supplemental information for this article can be found online at http://dx.doi.org/10.7717/peerj.15058#supplemental-information.

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
