# Peer review of "Identification of immune-related genes in acute myocardial infarction based on integrated bioinformatical methods and experimental verification"

_PeerJ, doi:10.7717/peerj.15058_

## Round 0.1 · original submission · Major Revisions

Dear Dr. Guo

Thank you for submission of your manuscript titled "Identification of immune-related genes in acute myocardial infarction based on integrated bioinformatical methods and experimental verification". This manuscript can be considered again for further revision provided valid justifications for the comments made by the reviewers.

Will look forward to the revised version of the manuscript.

Sincerely,
Dr. Mahesh Gokara

Reviewer 1 ·

Basic reporting

In accordance with criteria defined for publication in PeerJ, the manuscript lacks in relevant literature referencing and general English language needs improvement. The authors have a habit of writing long sentences with too many conjunctions which make it difficult to read and understand the essence of the study.

In terms of the known literature for immunological regulatory mechanism for AMI, the authors have claimed throughout the introduction and discussion sections that very little knowledge is available. Upon a simple search through PubMed I could identify various papers and reviews which have addressed the issue and have not been cited by the authors to define what part of the mechanism is still unclear and how their study is filling the gap.
A few links to the reviews/papers I found relevant and were not included in this study:
1. https://www.frontiersin.org/articles/10.3389/fimmu.2021.664457/full
2. https://www.ncbi.nlm.nih.gov/pmc/articles/PMC9174900/
3. https://www.ahajournals.org/doi/10.1161/CIRCRESAHA.121.318005

Can the authors explain via their response why this literature was not included? If these studies are relevant to the field, it is important the authors consider rewriting the introduction and discussion section. Claiming that “inflammatory regulation mechanism in AMI is not very clear” can only be justified if the authors clearly build their introduction around the inflammatory stages defined for a cardiac event and which stage of the event (pre event, cardiac event or post event) is the basis of their study.
From the experimental validation conducted, it seems to me that the authors are interested in identifying the markers when an AMI event has already happened but also state that such biomarkers are already available such as CK-MB, cTnT and cTnI.
It is true that not all mechanisms are clear for the overall immune related prognosis of the disease. But understanding that will require time series dependent analysis which is not being analyzed in this study.
Can the authors comment in terms of fold changes if they see the three known injury markers in the GEO datasets in comparison to the Hub genes identified by this study? This is important to establish the relevance of these Hub genes as biomarkers in comparison to ones already used in day-to-day practice.

Likewise, the result and discussion sections of the manuscript are also very ambiguous and not connecting the dots. The authors seem to just throw into words the results obtained through the bioinformatics analysis without making sense of the results with respect the literature available. For example, in results section, lines 210-212, 10 hub genes were identified. The authors neither in the results nor in the discussion section describe the relevance of these genes. Why did they see it in their analysis as hub genes and what information is available about then in the literature. The authors further state in lines 256-258 that a particular Hub gene MMP9 has already been established to be related to AMI but do not cite any reference? Also, would it not be of interest to the authors to see how the other hub genes are related to this MMP9 gene in terms of GO and KEGG analysis and fill in the gaps in the literature rather than just leaving the gene out in the further discussion?
Overall, the manuscript does hold some key findings, but the authors do not have a clear understanding of representing these results in a way that these may be used in future research to fill in the gaps in the immunological prognosis of AMI.
Some examples for illegible sections in the manuscript:
a. Abstract: “A total of 4,482 signiûcant up-regulated and 951 down-regulated genes were found in GSE66360 and GSE24519, 116 immune-related genes in close association with AMI were screened by WGCNA analysis, on the basis of GO and KEGG enrichment, these genes mostly clustered in the immune response, with construction of PPI network and LASSO regression analysis, this research found 3 hub genes among these differentially expressed genes. “
Contains too many “,” which can be broken into multiple sentences
b. Result: Result of immune cell infiltration: lines 220-246. The authors have just stated what they show in figure 6 in terms of words like positive correlation and negative correlation which do not add any value to the text. Either rewrite the complete section with relevant information or simply draw a clear table/heatmap to show the most important results. This long section makes the result obtained confusing and illegible.
c. Discussion: Lines 313- 317-- Each sentence starts with “Then we” which can be replaced by other more suitable phrases.
d. Lines 330-332—too many “and”
e. Various typos throughout the text.

Experimental design

Although the authors describe standard methods required to undertake any GEO based study, there are a few details not provided in the manuscript which I would request the authors to comment and include the responses clearly in the revised version of the manuscript.
a. The authors have selected GSE66360 and GSE24519 for conducting this study. Are these on only dataset available for AMI having control and patient data? If not, what was the criteria used to select only these two data sets?
b. The authors use STRING database to identify the Hub proteins taking the criteria of interaction pair score> 0.4. In the supplementary file provided (String.xlsx), the interaction score is the average of several independent scores involving neighborhood on chromosome, gene fusion, phylogenetics, co expression, experimentally determined interaction, database annotation and text mining. It seems as if the last two factors i.e. database annotation and text mining hold higher weightage in the overall score. I would request the authors to repeat this analysis by excluding the last two columns and consider an average of only column H (co expression) and I (experimentally defined interaction) or the first 5 columns to recalculate the interaction pair scores. Text mining is generally not used as a scientifically robust criteria to infer a protein-protein interaction. Hence it is important to know if the authors see the similar results even after removing the last two factors from their interaction pair scores.
c. Since the Protein-Protein interaction network is the source of LASSO and Hub gene identification, I will request the authors to repeat this analysis again if they find the results of the experiment suggested above (2b) change as per the defined criteria.
d. The authors use another dataset GSE97320 for validating their Hub gene which contains only 3 patients with AMI. What was the basis of selecting this dataset? Is this the only dataset available and how different it is from the ones selected in the initial pipeline?
e. The experimental validation through the mouse model looks interesting but what was the control used in the experiment? For any RT-PCR experiment, an internal control is required which would essentially be same for both groups (housekeeping genes). This is done to validate the experimental procedure itself. Please provide the details of the internal control and if the experiment was performed without it, please repeat, and reproduce the results.
f. Since most of the bioinformatic analysis is performed using R and its specific libraries, it is required by the authors to submit the codes used as a separate file to ensure reproducibility of the results. I request the authors to kindly provide the details of all the R packages used, with their version, the functionalities as well as parameters used to conduct the analysis.

Validity of the findings

Overall, the manuscript requires thorough revision both in terms of technical aspects as well as presentation of the new knowledge with respect to known literature.

Reviewer 2 ·

Basic reporting

Liu et al studied the immune-related genes in acute myocardial infarction using publicly available datasets and proposed validation using three genes in mice. However, this manuscript is not well written, lacks methodological details, and contain number of disconnected analyses. Background did not include sufficient details of the existing studies and in the results section author did not mention clearly what was known and what they have found. Here I commented on some of the issues but there are many.

Experimental design

The entire premise of the work is based on two publicly available independent datasets which showed very different results in the number of DEGs. Authors took union of DEGs of datasets and carried further analysis. Why do authors observe such a higher difference in the number of DEGs? Ideally, authors should take the union of the genes that are consistently differentially expressed between studies and are reproducible. Why did authors took union instead of intersection of DEGs?

This is important to show the quality, and batch effects of data where authors could use hierarchical clustering or, PCA of the samples. Those analyses could also provide a better understanding why these two datasets used in this study showed very different results in the number of DEGs.

In Fig 2D, there require separate venn diagrams for up and down-regulated genes.

This study uses array data (GSE66360 and GSE24519). However, in L165 and L166 authors mentioned “this study examined the RNA-seq data”!

L87 and L169: Two different p-values were mentioned, which one is correct? Why did the authors not use FDR corrected value instead of p-value? Multiple hypothesis testing must need to be performed in differential gene expression analysis.

In Figure 2D, E, why are authors showing this data? Top 10 genes in two studies are not even similar.

Figure 3B is uninterpretable because of illegible labels and complex color patterns. Can authors provide a simple and quantitative analysis of the data?

L188: “This module belongs to up-regulated genes, with a total of 132 genes (Fig. 3C)”. Why are authors referring to Figure 3C? How Figure 3C is telling about “This module belongs to up-regulated genes”?

L189: “We compared the immune gene list”- which list was used? Any citations?

In general, figure legends did not include sufficient details.

Validity of the findings

In Fig 8, author showed validation data where in GSE97320. SOCS2, FFAR2 and MYO10 were significantly upregulated in the AMI group than in the control group at P<0.01. Are those genes significantly differentially upregulated by Limma after FDR correction? Fig 8B, how many biological replicates were used for qPCR? Why did authors only look into only three genes - SOCS2, FFAR2 and MYO10? Why authors did not test other genes (e.g., MMP9) in the mouse study?

L266: “constructed mice myocardial infarction model” how this mouse model was created?

Additional comments

“Acute myocardial infarction (AMI) is still one of the main causes of death in the world”, please provide the statistics instead of saying “one of the main causes”.

May need grammatical corrections “Immune cell inûltration showed possible the immune system intervention points in the process of AMI.”

L87: this line needs to be rewritten.

L87: Why (Erhard 2018) is cited?

Fig 2 legend: “Wayne Diagram”?

---

## Round 0.2 · Major Revisions

The authors have Appealed and we have consulted with the Section Editor.

We are willing to receive a revised manuscript which must comprehensively describe how all the reviewer queries were dealt with. Authors should also provide a 'true' tracked changes document which shows the edits made between the first version and the most recent one as well as which references have been added

We thank you as well for agreeing to improve the language.

· Appeal

Appeal


· · Academic Editor

Reject

Dear Caixia Guo ,

Thank you for submitting your manuscript, "Identification of immune-related genes in acute myocardial infarction based on integrated bioinformatical methods and experimental verification," for reconsideration. After carefully reevaluating the reviewers' suggestions and valuable comments provided by the section editor, we have decided to reject the manuscript.

We understand that the review process can be difficult, and we appreciate the time and effort you have put into revising your manuscript. However, after considering the feedback from the reviewers and the section editor, we have determined that the manuscript is not ready for publication in our journal.

Thank you again for your submission, and we hope you will continue to consider PeerJ as a venue for your future research.
All the Best

Reviewer 1 ·

Basic reporting

The authors have improved on the literature citation and general English as suggested. Overall the manuscript has improved along with figures and discussion related to crucial findings.
There is still scope for minute improvement such as
Lines 48-63: please check tenses of the statement in this paragraph. Half of the paragraph is in present and other half in future.
Line 285-857: Please refer to MMP9 paper (Cheng, 2022) i.e. in the first occurrence of MMP9 and why authors are not following up with this gene in the future study.
The authors have noted 116 immune related genes in abstract and 118 in discussion. Please correct this error.

Experimental design

I am satisfied with the modifications in the experimental design as suggested previously.

Validity of the findings

I appreciate authors identifying a better dataset (GSE48060) for providing validation.

---

## Round 0.3 · accepted · Accept

I am writing to inform you that the manuscript you submitted to PeerJ titled ‘Identification of immune-related genes in acute myocardial infarction based on integrated bioinformatical methods and experimental verification’ has undergone a thorough review process, and I am pleased to inform you that it has been accepted for publication.

We would like to take this opportunity to thank you for choosing PeerJ as a venue for your research.

Once again, congratulations on your acceptance, and we look forward to publishing your work.

Best regards,
Mahesh Gokara